# The Formation Process and Strengthening Mechanism of SiC Nanowires in a Carbon-Coated Porous BN/Si_3_N_4_ Ceramic Joint

**DOI:** 10.3390/ma15041289

**Published:** 2022-02-09

**Authors:** Yanli Zhuang, Tiesong Lin, Peng He, Panpan Lin, Limin Dong, Ziwei Liu, Leiming Wang, Shuo Tian, Xinxin Jin

**Affiliations:** 1Heilongjiang Provincial Key Laboratory of CO_2_ Resource Utilization and Energy Catalytic Materials, School of Material Science and Chemical Engineering, Harbin University of Science and Technology, Harbin 150040, China; zhuangyanli@hrbust.edu.cn (Y.Z.); donglm@hrbust.edu.cn (L.D.); liuziwei199604@163.com (Z.L.); wangleiming199905@163.com (L.W.); tianshuo199606@163.com (S.T.); au.zn.zn@163.com (X.J.); 2State Key Laboratory of Advanced Welding and Joining, Harbin Institute of Technology, Harbin 150001, China; pplin@hit.edu.cn

**Keywords:** porous BN/Si_3_N_4_, SiC nanowires, formation process, strengthening mechanism

## Abstract

Porous BN/Si_3_N_4_ ceramics carbon-coated by carbon coating were joined with SiCo38 (wt. %) filler. The formation process and strengthening mechanism of silicon carbide nanowires to the joint were analyzed in detail. The outcome manifests that there is no distinct phase change in the porous BN/Si_3_N_4_ ceramic without carbon-coated joint. The highest joint strength was obtained at 1320 °C (~38 MPa). However, a larger number of silicon carbide nanowires were generated in the carbon-coated joints. The highest joint strength of the carbon-coated joint was ~89 MPa at 1340 °C. Specifically, silicon carbide nanowires were formed by the reaction of the carbon coated on the porous BN/Si_3_N_4_ ceramic with the SiCo38 filler via the Vapor-Liquid-Solid (VLS) method and established a bridge in the joint. It grows on the β-SiC (111) crystal plane and the interplanar spacing is 0.254 nm. It has a bamboo-like shape with a resemblance to alloy balls on the ends, and its surface is coated with SiO_2_. The improved carbon-coated porous BN/Si_3_N_4_ joint strength is possibly ascribed to the bridging of nanowires in the joint.

## 1. Introduction

BN/Si_3_N_4_ composite ceramics show wonderful thermal shock resistance, favorable chemical durability, and especially low and stable dielectric constants and dielectric losses over wide temperature ranges [1,2,3,4]. They have become one of the most promising wave transparent material candidates for high Mach number aircraft radomes [5,6]. In order to satisfy the requirements of broadband transmission of the components and to further reduce the dielectric constant at high temperature, materials are usually designed to be porous [7]. However, the intrinsic brittleness of ceramic materials and the limitations of preparation processes restrict their application in large-scale and complex-shaped components. To solve the above-mentioned drawbacks, joining technology is widely used in ceramic material assembly. For instance, brazing and diffusion welding are common methods used for bonding ceramic materials [8,9,10,11]. However, due to the high surface roughness of porous ceramics, they cannot meet the basic requirements of diffusion welding. In contrast, brazing is more suitable for porous ceramic joining.

At present, there are few reports on the direct brazing of porous ceramics. Porous Si_3_N_4_ ceramics are usually brazed to metals (such as TiAl and Invar). Ag-Cu-Ti fillers are mainly used in the joining process [12,13,14,15]. However, the joint strength is low on account of the large discrepancy in the physicochemical properties between the substrate and the filler metal. To improve the connection of the joints, we have previously reported a silicon-based filler for brazing porous Si_3_N_4_ ceramics [16,17]. Before bonding, porous BN/Si_3_N_4_ ceramics were preimpregnated with phenolic resin organic solution assisted by subsequent high-temperature treatment to prepare carbon coating on their surfaces, which could provide carbon source for the generation of SiC nanowires during the bonding process. The free Si in Si-Re (Re = Ti, Co) filler is used as silicon source, the transition group metal Re is used as catalyst, and the pores of the substrate can provide space for the growth of nanowires. During the brazing process, the carbon coating on the substrate and free Si in the solder react in situ under the catalytic action of Re, and SiC nanowires are formed at the interface. Results demonstrated that the bonding strength was greatly enhanced by pre-fabricating carbon coating on porous ceramics surface, resulting in the creation of SiC nanowires in the joint. However, the forming process and strengthening mechanism of SiC nanowires to the joint have not been clarified.

In this study, carbon-coated porous BN/Si_3_N_4_ ceramics are bonded with a SiCo38 alloy. The effects of bonding process on the joint morphology and mechanical properties are studied, whereby emphasis is placed on the formation process of silicon carbide nanowires and the strengthening mechanism of silicon carbide nanowires on the joint.

## 2. Material and Methods

The porous BN/Si_3_N_4_ ceramics fabricated by a non-pressure sintering method were provided by the Institute for Advanced Ceramics, Harbin Institute of Technology [18], and were mainly made up of Si_3_N_4_ and h-BN powders. The Al_2_O_3_ and Y_2_O_3_ powders were used as sintering aids. Poly (methyl methacrylate) (PMMA) microspheres with a diameter of ~10 μm were used as pore-forming agents. Before brazing, the base material was first cut with an inner circle slicer into test specimens with sizes of 5 × 5 × 5 (mm^3^) and 10 × 10 × 5 (mm^3^). The specimen surface was burnished making use of a 2000# diamond grinding disc and deep cleaned, ultrasonically assisted, in alcohol for 20 min. The porous BN/Si_3_N_4_ ceramic samples were subsequently submerged into a homogenous organic solution of methyl alcohol and phenolic resin (the volume ratio was 2:1) to undergo modification under a vacuum of 0.9 × 10^−1^ kPa for 15 min at room temperature. The carbon-coated specimens were successively placed in an air oven at 60, 120, and 110 °C for one hour, and at 160 °C for 2 h after that. Then, the specimens were put in a vacuum sintering furnace for pyrogenic decomposition at 1100 °C for 15 min under a vacuum of 1.5 × 10^−2^ Pa. The micrographs of reference and carbon-coated porous BN/Si_3_N_4_ ceramic specimens are shown in Figure 1. It can be seen from Figure 1b that the carbon layer has a coating on the ceramic’s surface after modification. This has been identified as a carbon layer in previous studies [17]. From Figure 1c can be seen that the corresponding thickness of the carbon layer is about 3 μm when the volume ratio of methyl alcohol and phenolic resin is 2:1.

The filler metal’s composition used in this paper was SiCo38 (wt. %) [19]. The micrographs of SiCo38 alloy specimen is shown in Figure 1d. The preparation method and phase composition of filler have been reported in detail in previous studies [16]. During the welding, the base material and filler are placed with sandwich manner in a vacuum furnace. The temperature was selected as 1300 °C (about 30 °C higher than the filler material melting point). The target peak temperature had subsequently increased to 1320, 1340, and 1360 °C, in 20 °C temperature intervals, respectively. The “sandwich” specimens were heated to the bonding temperatures with a 10 °C/min rate and held at welding temperature for 10 min. After brazing, the specimens were cooled down to 200 °C at 5 °C/min. The equipment used for phase analysis, microstructure observation, and mechanical property analysis of the joint was the same as previously reported [16].

## 3. Results and Discussion

### 3.1. Microstructure of the Porous BN/Si_3_N_4_ Joint Domain

The SiCo38 alloy was applied to weld the porous BN/Si_3_N_4_ ceramic at 1320 °C for 10 min. Figure 2 illustrates the joint micrographs. The joint areas were homogeneous and no obvious new phase was observed. Figure 3 reveals the X-ray diffraction (XRD) results of the joint and it can be concluded that Si_3_N_4_, BN, CoSi_2_, and Si are present in the joint. Energy spectrum analysis (EDS) of the corresponding in Figure 2b was carried out and the results are given in Table 1. From the combination of the XRD and EDS data, it can be deduced that the phases of the A, B, and C locations in Figure 2b are CoSi_2_, Si, and Si_3_N_4_, respectively. The above results indicate that no new phase is formed in the joint.

Figure 4 reveals the morphologies of the reference welded joints bonded with the SiCo38 filler at different temperatures. The width of the weld reduces from 35 to 18 μm when the temperature rises from 1300 to 1360 °C. In addition, when the temperature is higher than 1340 °C, CoSi_2_ aggregates and grows upwards, giving rise to an unevenness of the weld composition. The variation of the weld width may be influenced by the viscosity of the filler. In general, the viscosity of the filler metal decreases with the temperature increase. The decrease of the viscosity promotes the wetting and spreading of the filler. Therefore, more filler metal is extruded from the weld at higher temperatures, resulting in the width of the weld becoming narrower after cooling.

Figure 5 presents an influence of welding temperature on the joint strength. Within the range of the joining temperature, the joint strength increases first and then decreases gradually. The highest strength was obtained at 1320 °C (~38 MPa). Figure 6 shows SEM images of the typical fracture microstructure of the joint. It is observed from Figure 6a that the breakage mainly takes place on the interface between the substrate and the solder metal. A small amount of solder can be observed on the base material surface. The fracture is relatively flat, which is a typical brittle fracture. In the magnified of region “1” (Figure 6b), cracks and a small number of ceramic particles are observed on the surface of the filler metal. Figure 6c indicates that the substrate surface at the fracture is also rough and uneven. It is possible that the liquid filler infiltrated the pores of the substrate during the joining process, giving rise to a nailing effect. During the shearing process, a part of the ceramic particles was pulled off at the interface.

The joint strength can mainly be affected by the interface bonding force, weld width, and phase distribution. When the temperature is 1300 °C, the viscosity of the filler metal is relatively high. Because there is no chemical reaction at the interface, the filler is mainly connected to the substrate by the nailing effect. The high viscosity of the filler leads to poor wetting and spreading; therefore, the nailing effect between the substrate and the filler metal is weak, which makes the interface bonding force weak. In addition, the width of the weld is larger at a lower temperature (exhibited in the Figure 4a). The large weld may not be conducive to improving the joint strength [20], hence, resulting in a lower joint strength.

When the temperature increases, the viscosity of the filler decreases, and the fluidity increases. Under the action of capillary forces, the infiltration of the filler metal into the substrate pores is enhanced, and so the nailing effect at the interface is also enhanced. In addition, the width of the weld is gradually narrowed. The combination of these factors promotes to strengthen the joint shear strength. However, when the temperature is higher than 1340 °C, CoSi_2_ aggregates and grows upwards due to the uneven phase distribution in the weld, resulting in a sharp decrease in the joint strength.

### 3.2. Microstructure of the Carbon-Coated Porous BN/Si_3_N_4_ Joint Domain

Figure 7 shows the appearance characteristics of the carbon-coated porous BN/Si_3_N_4_ ceramic joints by carbon coating bonded with the SiCo38 alloy at 1340 °C for 10 min. Figure 7a,c shows that the joint is connected well, and no obvious holes or cracks are observed at the interface. The other one remarkable phenomena of the joint is the appearance of nanowires in the pores at the interface. The nanowires possess a ball at the ends (as shown in Figure 7b). In order to determine the reaction product in the joint, XRD and EDS experiments were undertaken and the test results are given in Figure 8 and Table 2. The XRD results in Figure 8 show peaks at 2θ = 36°, 41°, 60°, and 72° resulting from silicon carbide. This demonstrates that a reaction product is formed in the joint.

The energy spectrum analysis of the nanowires (as shown in point A in Figure 7b) demonstrates that A is composed of Si, Co, N, B, and C, where the abundances of Si, N, and C are relatively high, and the atomic ratio of Si and C elements is close to 1:1. This indicates that the nanowires may be SiC. This conclusion will be demonstrated in the follow-up mechanism section. The combination of EDS and XRD analysis results infers that SiC particles may come into being in the joint (See point B in Figure 7c). In order to qualitatively analyze the phase composition in the joint, the transmission electron microscope (TEM) analysis was performed on the area marked with a white box in Figure 7c. Figure 9 presents the element mapping analysis on the area marked with a red box in Figure 7c. It can be seen that area “1” is dominated by Si and C, area “2” is dominated by Si and Co, and area “3” is dominated by Si, B, and N. Figure 10 shows the SAED patterns of “1”, “2”, and “3” marked in Figure 9a. The analysis results can determine that area “1” is SiC, area “2” is CoSi_2_, and area “3” is Si. From the element mapping (Figure 9) and the SAED patterns (Figure 10) of the products, in combination with the EDS and XRD results, it can be proved that SiC particles were generated in the joint. According to the above research, SiC particles were generated by the reaction of the carbon-coated on the porous BN/Si_3_N_4_ ceramic with the liquid silicon of the SiCo38 alloy [14,15]. In the following section, the crystal structure and formation process of the SiC nanowires will be discussed in detail.

### 3.3. Formation Process of SiC Nanowires

The most common growth methods of SiC nanowires are vapor-solid (VS), vapor-liquid-solid (VLS), and liquid-liquid-solid (LLS) [21,22,23,24]. Among them, there are similarities between the VS and VLS methods, being that the original components forming the nanowires are all gaseous at high temperatures. However, the difference is that the VLS method usually requires transition metals (such as Ni, Fe, or Co) as catalysts. Furthermore, the nanowires grown by the VLS method exhibit balls at their tips, which is a unique characteristic of this method. By contrast, the liquid-liquid-solid method usually employs low melting point metals (such as In or Sn) as catalysts, and the target product is formed from the decomposition of the precursor of organometallic compounds.

For purpose of clarifying the crystal structure and formation process of the nanowires, transmission analysis was employed and the corresponding results are given in Figure 11. Figure 11a manifests that the nanowires have a diameter of 50–70 nm. They exhibit an alloy ball at the top (as shown in point “A”), which is a typical VLS feature. The EDS analysis results of the nanowires at different regions are shown in Table 3. The results indicate that point “A” mainly contains Si, Cu, Pt, and Co, and point “B” mainly contains Si, C, and Cu. Among these, Cu and Pt are elements originating from the sample support and gold spraying in the TEM sample. That is to say, the top ball of the nanowire is mainly made of Si and Co, and the backbone region is mainly made of Si and C. In addition, from Figure 11a it can also be seen that the growth orientation of the nanowires is deflected (as shown with arrow “1”) and the diameter is not uniform, but rather “bamboo-like” in shape (as shown with arrows “2” and “3”). Figure 11b shows the SAED patterns of the nanowires. It can be seen that the electron diffraction pattern is polycrystalline diffraction ring. By measuring the spacing of other planes in the HRTEM image (Figure 11b) and observing the angle between the planes, it is confirmed that it is 3C-SiC. It can be calibrated by cubic β-SiC diffraction pattern with electron beam incident direction $\;\left[0\;1\;\bar{1}\right]$. From Figure 11b and c, it also can be observed that the surfaces of the nanowires are covered with a layer of amorphous material. According to literature, this amorphous material is mainly SiO_2_ [25,26]. The interplanar crystal spacing of the nanowires in Figure 11d is 0.254 nm, which corresponds to the β-SiC (111) crystal plane.

The above analysis shows that SiC nanowires are grown using the VLS method during the joining. This suggests that the original components of the SiC nanowires are gaseous and transition metals are present as catalysts during the brazing process. In other words, in the system consisting of the SiCo38 filler and the carbon-coated porous BN/Si_3_N_4_ ceramic, solid carbon and liquid silicon are first converted into gases, and cobalt in the filler plays a catalytic role in the synthesis of the nanowires. Therefore, the formation of the nanowires may consist of the following four parts (as shown in Figure 12) [21,27]. The first part is the formation of CO and SiO gases. They may be formed by the reaction of oxygen in the system with the carbon layer and the liquid silicon, respectively (as shown in Equations (1) and (2)). The second part is the formation of the SiC crystal nucleus, which is likely to be formed by the reaction of liquid silicon or gaseous SiO with the carbon layer (as shown in Equations (3) and (4)). The third part is the formation of catalyst droplets. This is mainly because the SiC crystal nucleus gradually diffuses from a high concentration region to a low concentration region, and finally precipitates on the surface of the liquid filler. In this process, the liquid filler is pushed out onto the surface of the crystal nucleus, and finally forms catalyst droplets on the surface. The gaseous CO and SiO in the system then continue to dissolve into the catalyst droplets, and further react to form a new SiC crystal nucleus. When the SiC crystal nucleus in the droplets reaches saturation, it precipitates from the solid-liquid interface and grows into nanowires along the one-dimensional direction (as shown in Equations (5) and (6)).

In order to verify the feasibility of the above reactions, the thermodynamic calculation of Equations (1) to (6) was carried out by the software HSC Chemistry. The results are displayed in Figure 13. The Gibbs free energy of Equations (1)–(4) and (6) are all less than zero when the temperature rises from 1280 to 1420 °C, indicating that these reactions can be spontaneous in the brazing process. However, the Gibbs free energy of Equation (5) is greater than zero, and so it is not a spontaneous process. Therefore, combined with the transmission analysis results in Figure 11, it can be assumed that SiC nanowires are in fact generated according to Equation (6).

There are two possible reasons for the deflection of nanowires during growth. On the one hand, a dislocation may form during the growth of the nanowires and the lattice is distorted, resulting in epitaxial growth. On the other hand, SiO_2_ is formed during the growth of the nanowires, and when the SiC crystal nucleus is constantly precipitated at the solid–liquid interface, its lattice arrangement may be strongly disturbed by SiO_2_, causing the catalyst droplet to deflect in the direction of local movement [28,29]. The diameter of the nanowires is mainly affected by the diameter of the catalyst droplet, while the diameter of the catalyst droplet is affected by vapor pressure [30,31]. Therefore, the reason that the SiC nanowires are bamboo-like in shape may be due to the differences between the dissolution rate of the CO and SiO gases and the precipitation rate of the SiC crystal nucleus in the catalyst droplet. When the dissolution rate of the gas is greater than the rate of crystallization, the diameter of the catalyst droplet increases, and the diameter of the nanowire increases correspondingly. However, when the dissolution rate is lower than the crystallization rate, the diameter of nanowires decreases correspondingly.
(1)$$C_{\left(s\right)}{+O}_{\left(g\right)}{=\mathrm{CO}}_{\left(g\right)}$$
(2)$${\mathrm{Si}}_{\left(l\right)}{+O}_{\left(g\right)}{=\mathrm{SiO}}_{\left(g\right)}$$
(3)$${\mathrm{Si}}_{\left(l\right)}{+C}_{\left(s\right)}{=\mathrm{SiC}}_{\left(s\right)}$$
(4)$${\mathrm{SiO}}_{\left(g\right)}{+2C}_{\left(s\right)}{=\mathrm{SiC}}_{\left(s\right)}{+\mathrm{CO}}_{\left(g\right)}$$
(5)$${\mathrm{SiO}}_{\left(g\right)}{+3\mathrm{CO}}_{\left(g\right)}{=\mathrm{SiC}}_{\left(s\right)}{+2\mathrm{CO}}_{2(g)}$$
(6)$${3\mathrm{SiO}}_{\left(g\right)}{+\mathrm{CO}}_{\left(g\right)}{=\mathrm{SiC}}_{\left(s\right)}{+2\mathrm{SiO}}_{2(s)}$$

### 3.4. The Effect of Brazing Temperature on the Interfacial Microstructure and Joint Strength

Figure 14 shows the joint morphology at different temperatures. It is observed that the interface was brazed well, and the weld width decreased from 34 to 18 μm as the temperature increased from 1300 to 1360 °C. Moreover, there is no obvious difference in the joint morphology at different temperatures. The variation of the weld width can be affected by two factors. Firstly, the temperature increase will promote the wetting and spreading of the filler. This enables the filler to be easily squeezed out of the weld at high temperature, leading to the reduction of the weld width. Secondly, the diffusion rate and reactivity of the atoms at the interface increase gradually as the temperature rises, which can enhance the reaction of the filler at the porous ceramic interface.

Figure 15 shows the influence of welding temperature on the joint strength of the carbon-coated porous BN/Si_3_N_4_ joints. The corresponding fracture morphologies of the joints are given in Figure 16. From Figure 15 can be observed that the joint strength increases gradually from 28 to 89 MPa when the temperature rises from 1300 to 1340 °C. However, the joint strength decreases to 77 MPa when the temperature continues to rise to 1360 °C. From the fracture morphology of the joints, it can be seen that the fractures mainly occurred at the interface, indicating that the interface was a welding weak area. Figure 17 shows the form of connection of the SiC nanowires in the joints. It can be observed from Figure 17 that the SiC nanowires with different directions may form a bridging connection during the growth process. From Figure 18 can be seen that a large number of nanowires have been pulled off during shearing. The SiC nanowires generated at the interface may affect the joint strength. When the brazing temperature is low, the SiC nanowires are short and there are few (as shown in Figure 16e). Moreover, most of them only grow in the pores without forming an effective bridge between the substrate and the filler. Hence, the interface adhesion force is weak, and the joint strength is low. When the temperature rises from 1320 to 1340 °C, the length of SiC nanowires increase gradually (as shown in Figure 16f,g), which can consume energy during the pulling out in the process of shearing. Therefore, the joint strength gradually increases. However, the nanowires coarsen and deform as the temperature continues to rise (as shown in Figure 16h) and the joint strength decreases. This may be due to the fact that SiC tends to grow into other types of nanowires at higher thermodynamic temperatures, such as 4H, 6H, since this stacking error will be more common and thus this will lead to the formation of malformed, mechanically weak nanowires. In brief, the origin of the increase and decrease of joint strength could be linked to the morphology and density of SiC nanowires. More detailed and in-depth studies are needed to confirm how they affect the joint strength.

## 4. Conclusions

In this paper, the microstructure and connection strength of a porous BN/Si_3_N_4_ ceramic joints bonded by a SiCo38 alloy were studied. The results showed no distinct phase change to the porous BN/Si_3_N_4_ part of the ceramic joint. However, a large number of nanowires were generated in the carbon-coated welded joints. The joint strength was also significantly improved. Hence, the formation process and strengthening mechanism of SiC nanowires to joints were analyzed in detail. The results showed that SiC nanowires were generated via the vapor-liquid-solid (VLS) method. The nanowires grew on the β-SiC (111) crystal plane and had an interplanar spacing of 0.254 nm. They were bamboo-like in shape, possessing an alloy ball at the ends, and their surface was coated with SiO_2_. The joint strength was affected by the joining temperature. Compared with the uncoated porous BN/Si_3_N_4_ ceramic joint (~38 MPa at 1320 °C), the highest joint strength of the carbon-coated BN/Si_3_N_4_ joint was ~89 MPa when the brazing temperature was 1340 °C. That is, the joint strength increased by ~134% after carbon coating. The improved carbon-coated porous BN/Si_3_N_4_ joint strength is possibly ascribed to the bridging of nanowires in the joint.

## Figures and Tables

**Figure 1 materials-15-01289-f001:**
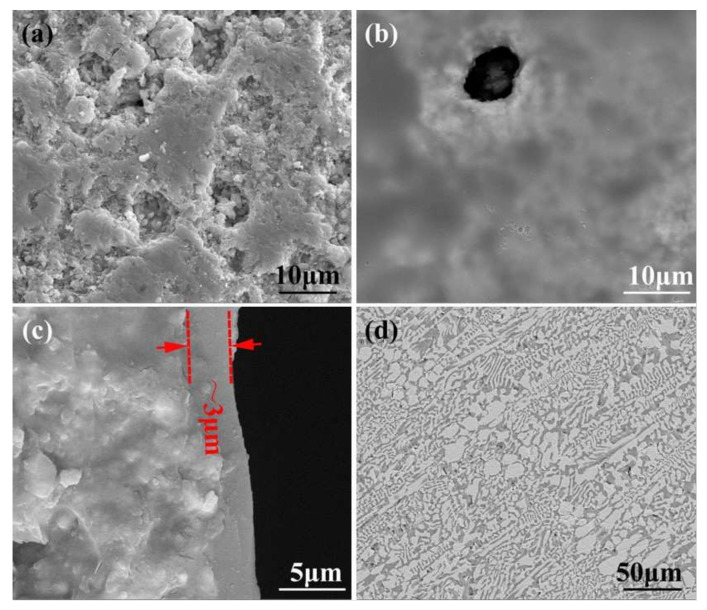
SEM micrographs of the reference (**a**) and carbon-coated (**b**) porous BN/Si_3_N_4_ ceramic specimens, thickness of carbon coating (**c**), and SiCo38 alloy (**d**).

**Figure 2 materials-15-01289-f002:**
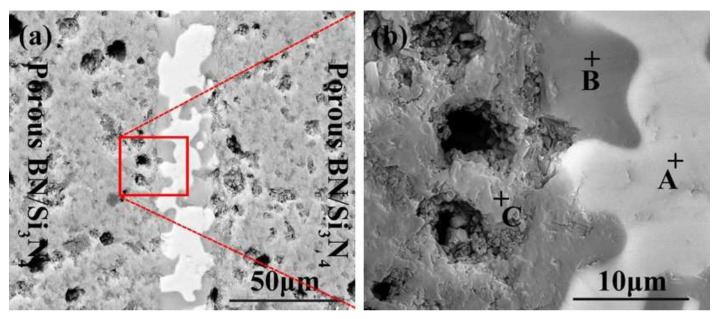
SEM micrographs of the reference welded joint brazed with SiCo38 alloy at 1320 °C for 10 min (**a**), and (**b**) is a magnified surface marked in (**a**).

**Figure 3 materials-15-01289-f003:**
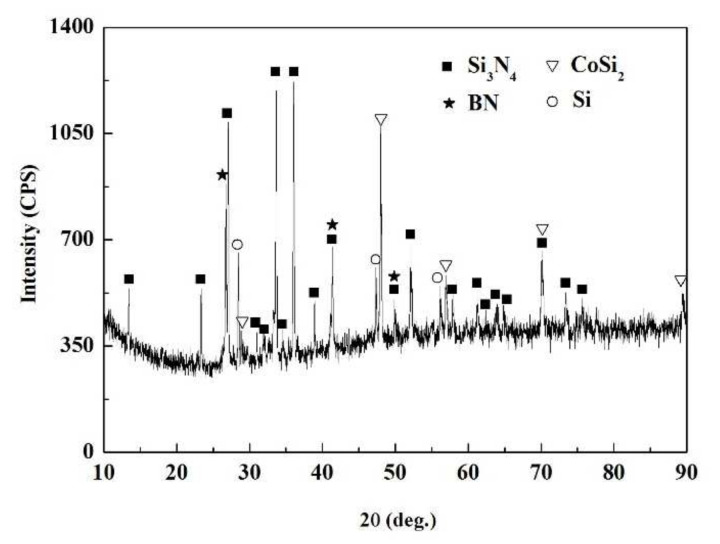
XRD patterns of the fractured surface of the reference welded joint.

**Figure 4 materials-15-01289-f004:**
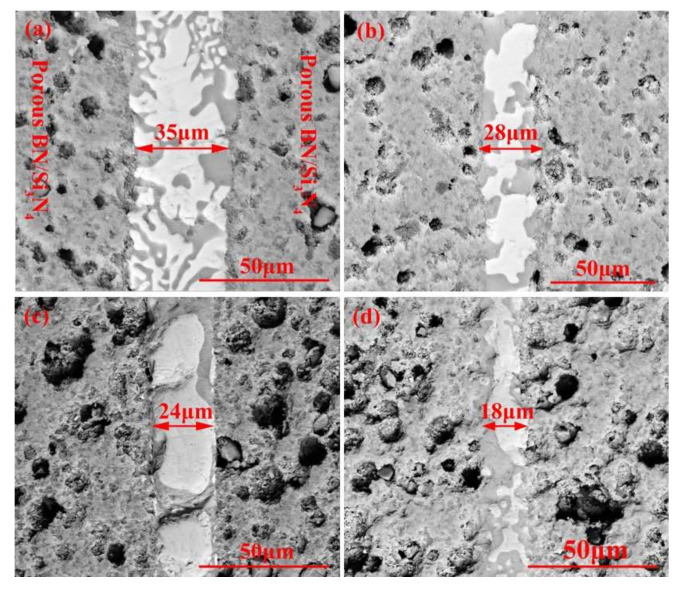
SEM micrographs showing reference welded joint brazed with SiCo38 alloy for 10 min at: (**a**) 1300 °C, (**b**) 1320 °C, (**c**) 1340 °C, and (**d**) 1360 °C.

**Figure 5 materials-15-01289-f005:**
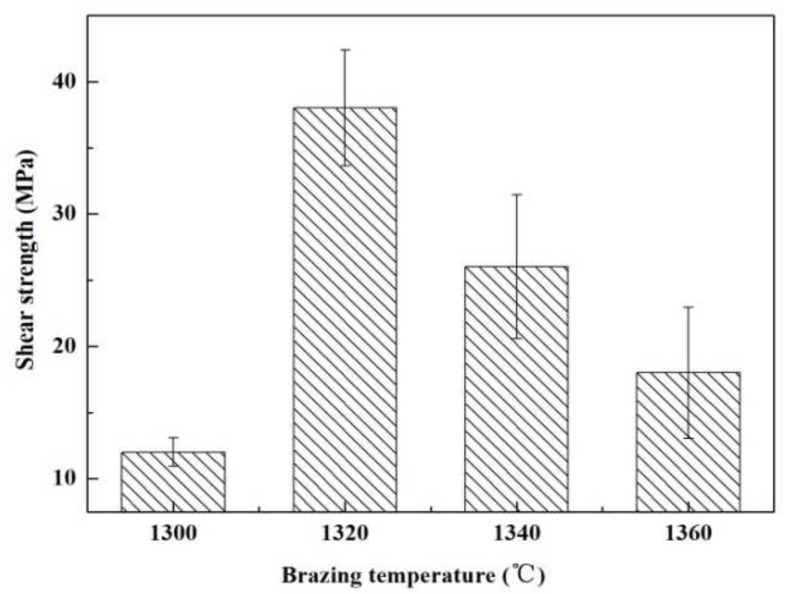
The temperatures impact on the reference welded joint strength bonded with the SiCo38 alloy for 10 min.

**Figure 6 materials-15-01289-f006:**
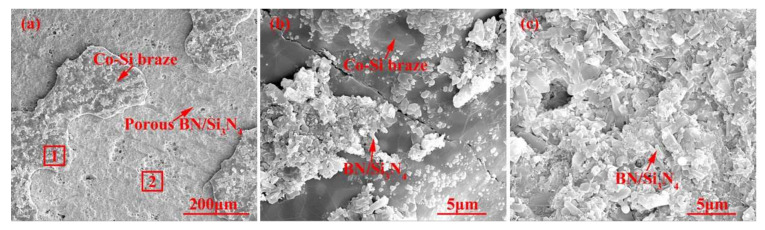
(**a**) Appearance characteristics of the reference welded joint fracture bonded at 1320 °C for 10 min, (**b**) Area “1” of (**a**) magnified, and (**c**) Area “2” of (**a**) magnified.

**Figure 7 materials-15-01289-f007:**
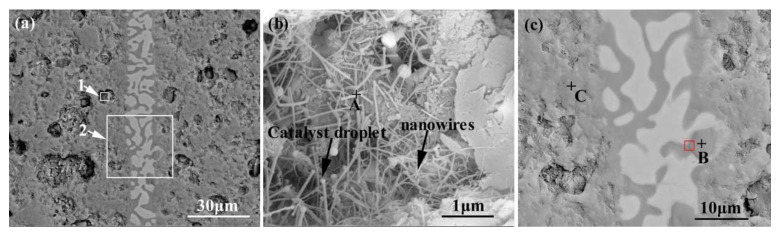
(**a**) Appearance characteristics of the carbon-coated welded joint bonded with SiCo38 alloy at 1340 °C for 10 min, (**b**) Area “1” of (**a**) magnified, and (**c**) Area “2” of (**a**) magnified.

**Figure 8 materials-15-01289-f008:**
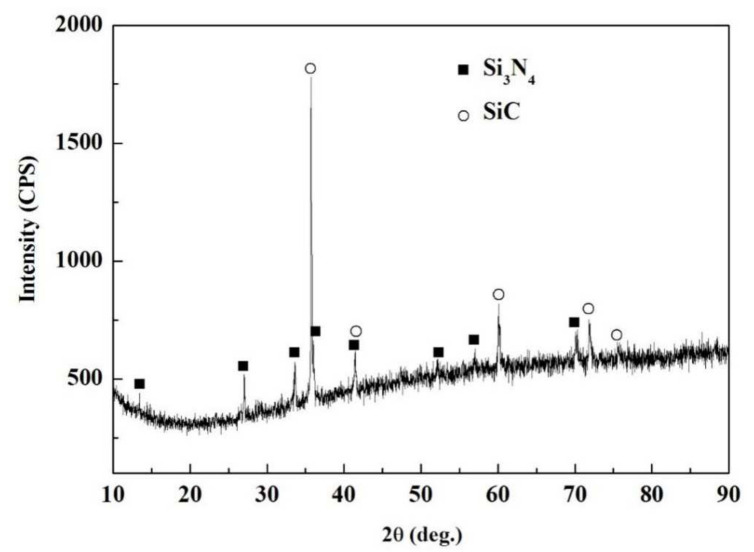
XRD patterns of the carbon-coated welded joint bonded with the SiCo38 alloy at 1340 °C for 10 min.

**Figure 9 materials-15-01289-f009:**
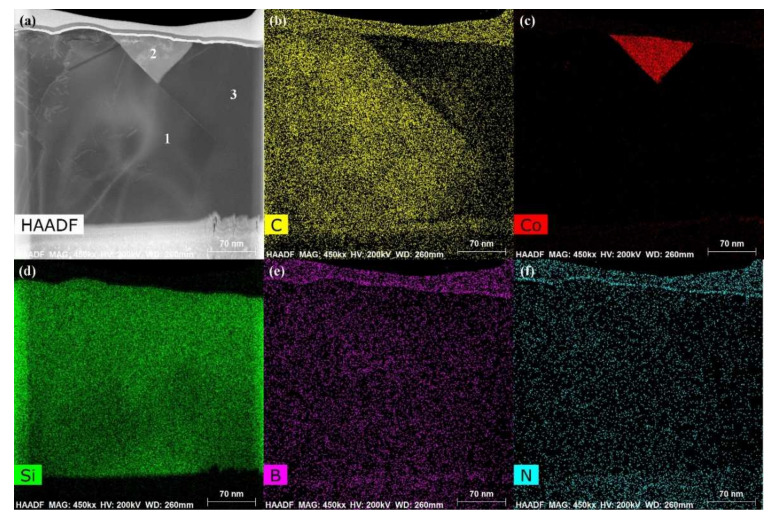
TEM images and the elements analysis on the area marked with a white box in Figure 7c: (**a**) Bright field image, (**b**–**f**) element analysis of Si, C, Co, B, and N.

**Figure 10 materials-15-01289-f010:**
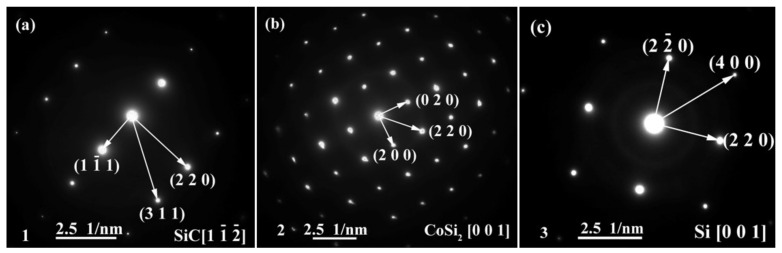
SAED patterns of “1”, “2”, and “3” marked in Figure 9a showing: (**a**) SiC, (**b**) CoSi_2_, and (**c**) Si.

**Figure 11 materials-15-01289-f011:**
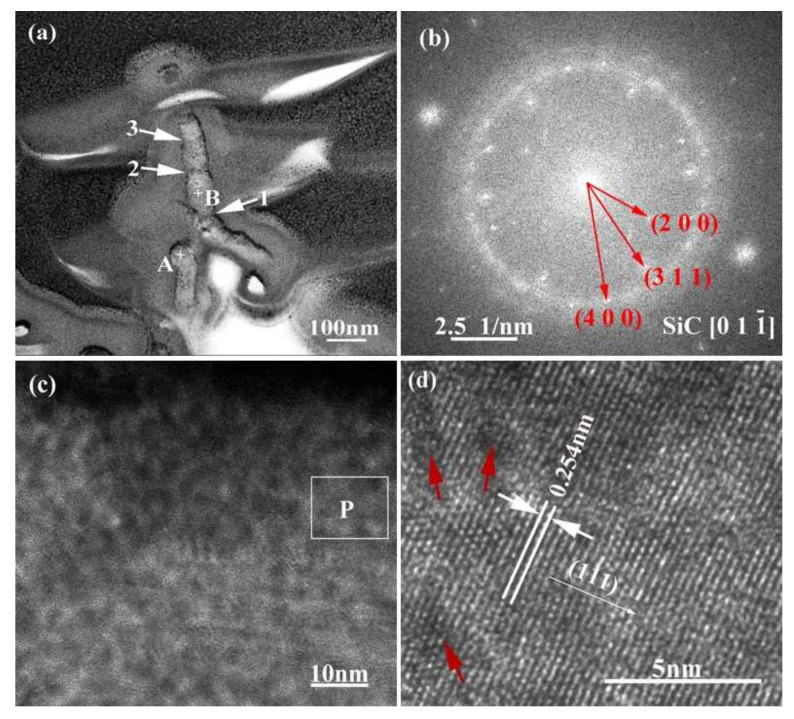
TEM image (**a**) and SAED pattern (**b**) of SiC nanowires, where (**c**) is HRTEM image of area “B” labelled in (**a**), and (**d**) is a magnified HRTEM image of area “P” labelled in (**c**).

**Figure 12 materials-15-01289-f012:**
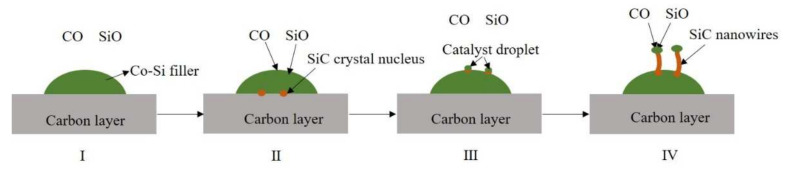
Schematic diagram of SiC nanowire formation.

**Figure 13 materials-15-01289-f013:**
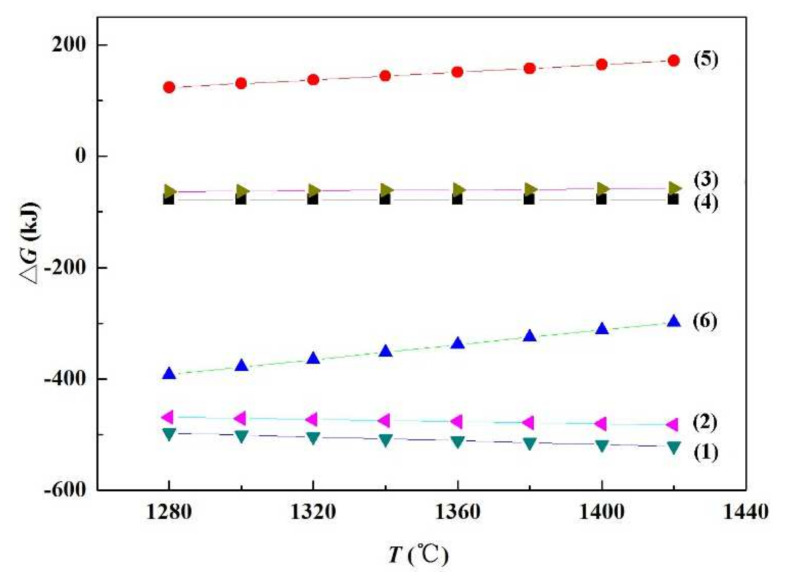
Gibbs free energy for each reaction equation at different temperatures.

**Figure 14 materials-15-01289-f014:**
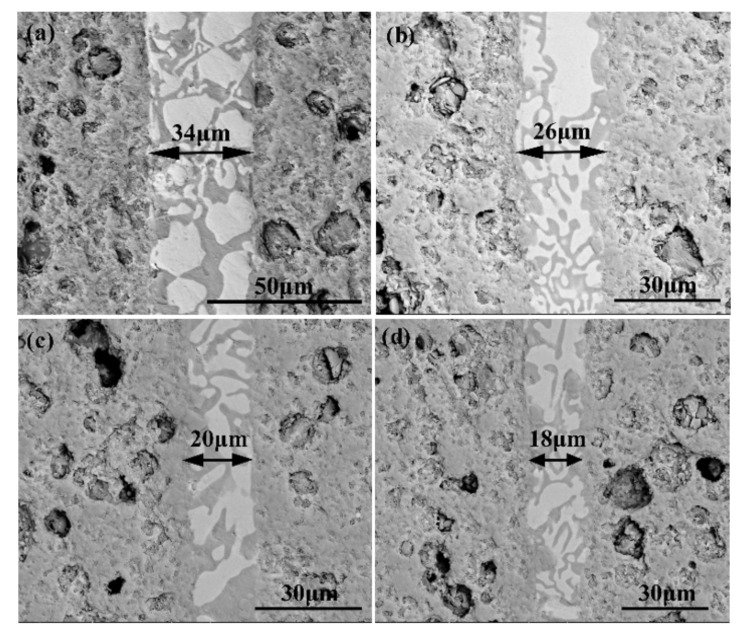
Microstructures of the carbon-coated welded joints bonded at different bonding temperatures for 10 min: (**a**) 1300 °C, (**b**) 1320 °C, (**c**) 1340 °C, (**d**) 1360 °C.

**Figure 15 materials-15-01289-f015:**
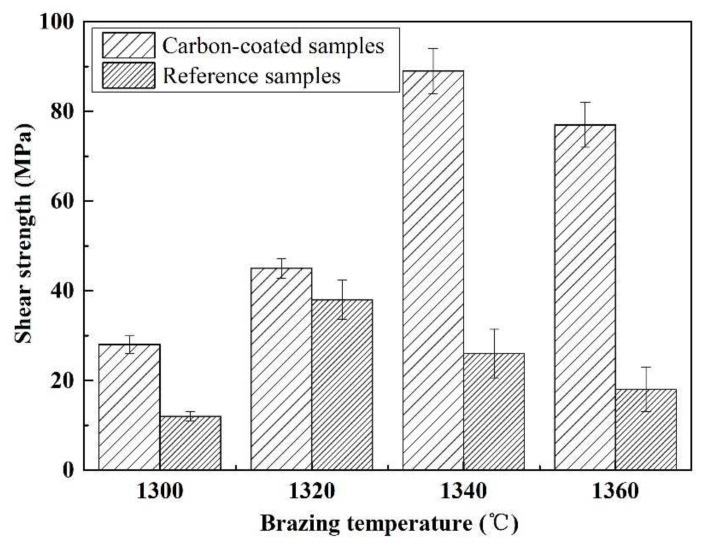
The temperatures impact on the carbon-coated welded joint strength bonded with the SiCo38 alloy for 10 min.

**Figure 16 materials-15-01289-f016:**
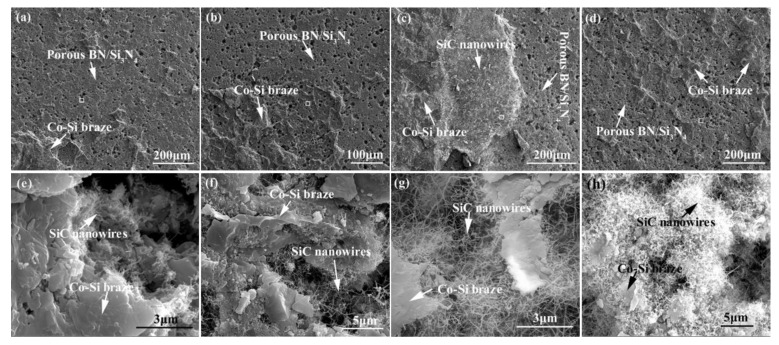
Appearance characteristics of the carbon-coated welded joint fracture bonded at different temperatures for 10 min: (**a**) 1300 °C, (**b**) 1320 °C, (**c**) 1340 °C, (**d**) 1360 °C, (**e**–**h**) Magnified white boxed area of (**a**), (**b**), (**c**), and (**d**).

**Figure 17 materials-15-01289-f017:**
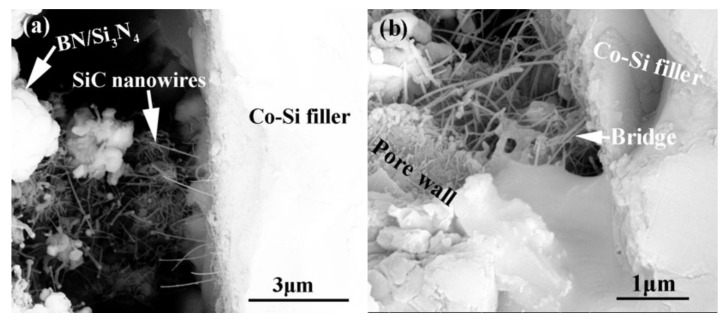
Connection form of the SiC nanowires in the joints. (**a**) SiC nanowires with different directions, (**b**) the formed bridge.

**Figure 18 materials-15-01289-f018:**
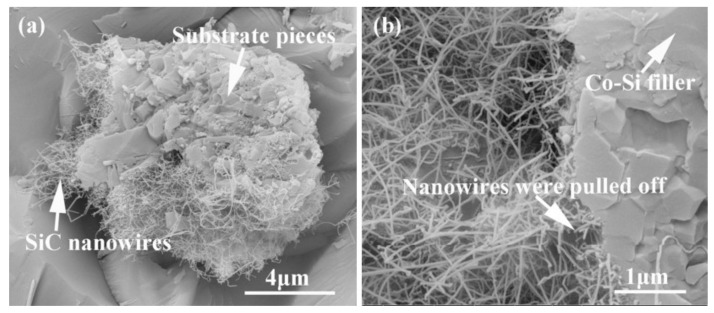
SiC nanowires were pulled off during the shear test from samples brazed at 1340 °C for 10 min. (**a**) Pieces of substrate were cut off, (**b**) pieces of the filler were cut off.

**Table 1 materials-15-01289-t001:** The EDS results of the points signed in Figure 2b.

Position	Composition (at. %)				Possible Phases
	Si	Co	N	B	
A	59.02	27.35	8.90	4.73	CoSi_2_
B	80.80	1.47	9.72	8.01	Si
C	51.29	2.51	31.95	14.25	Si_3_N_4_

**Table 2 materials-15-01289-t002:** The EDS results of the points signed in Figure 7b,c.

Position	Composition (at. %)					Possible Phases
	Si	Co	N	B	C	
A	38.41	0.57	25.91	5.01	30.10	SiC
B	42.71	0.49	15.43	4.44	36.93	SiC
C	38.60	2.05	20.89	16.01	22.45	SiC, Si_3_N_4_

**Table 3 materials-15-01289-t003:** The EDS results of the points marked in Figure 11a.

Position	Composition (at. %)						
	Si	C	N	O	Cu	Pt	Co
A	30.25	8.36	0.2	1.66	30.36	9.39	19.78
B	36.13	25.91	1.20	2.78	22.96	8.02	3.00

## Data Availability

The data presented in this study are available on request from the corresponding author.

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
