# Peer review of "The Formation Process and Strengthening Mechanism of SiC Nanowires in a Carbon-Coated Porous BN/Si3N4 Ceramic Joint"

_materials, 2022, doi:10.3390/ma15041289_

Round 1

Reviewer 1 Report

Review on “the formation process and strengthening mechanism of SiC nanowires in a carbon-coated porous BN/Si3N4 ceramic joint” by Yanli Zhuang , Tiesong Lin , Peng He 2, Panpan Lin, Limin Dong , Ziwei Liu , Leiming Wang , Shuo Tian and Xinxin Jin.

The manuscript review the structural and strengthening change of a annealed ceramic joint. The authors claims that the formation of SiC nanowire is the cause of the improved mechanical properties. Overall the manuscript is well written.

I have some comment detailed below, tentatively ascribed Major/Minor. My main problem in the manuscript is in the final section (see the “Really Major” comment).

The rest is more about presenting and organising the data.

* [Major] The introduction section is well organised and clearly set up the interest of the proposed research work.

However the main interest of the manuscript, the strengthening of the joint using a Carbon coating is not introduced at this moment.

→ the author need to detail their methodology somewhere, possibly at the end of the intro, stating that they will study the structural and mechanical effect of the carbon coating on the BN/Si3N4.

(see also next comment on the material & method)

* [Major] The materials and method is clear and detailed. However there are 2 majors points to be addressed :

- a description of the carbon coating is missing in this section.

- a schematic of the structure of each sample is missing, separing the one with and without carbon coating, and describing the thickness and lateral size of each element (BN/Si3N4, CoSi, BN/Si3N4… and carbon)

* [Minor] Fig1, all the white text cannot be read against the light grey SEM. Please use an alternative color with more contrast with the background. Also separate a little the (a) and (b) SEM for clarity.

* [Minor] Fig3, same comment as Fig1 for all the white text and marks ( including SEM scale). Please use an alternative color with more contrast with the background

* [Minor] Fig5, white text and marks , same as Fig3.

* [Major] at page 5, line 139. Here should begin a new subsection with a title such as “3.2 Microstructure of the carbon-coated porous BN/Si 3N4 joint domain”. Otherwise the 3.1 section is far too large and the comparison between standard and carbon-coated joint is not directly clear to the reader

* [Major] the modified / unmodified naming scheme is not clear enough to the reader. Please use explicit naming such as “reference” and “carbon-coated” to describe the two type of sample in the manuscript.

*[Minor] Fig6, please indicate the TEM area also in the legend of Fig6(c), use a different color if possible (white line is too thin, not visible enough)

*[Major] while not complicated the Fig8 and Fig9 are barely described in the main text (see line 158 & 159) and the reader is left to conclude him/herself. Please detail each figure and state clearly what conclusion can be drawn.

*[Minor] Why is there no STEM-EDX map of Fig10 (similarly to Fig 8?) ? In particular, the Oxygen map would help the author in verifying the presence of SiO (and other oxides).

*[Major] section 3.2, what is the oxygen source ? Why the the carbon-free sample not affected by oxygen diffusion ? (SiOx / CoOx )

*[Major] the chemistry (line 255 and up) involves substantial amount of oxygen assisted transport of Si and C. Can such large amount of oxygen also affect the other component of the joints (Co, BN, Si3N4) ?

[Minor] : please add the reference (carbon free) data to the Fig14 in addition to the carbon-coated sample (use another color), so the reader can appreciate directly the difference with / without carbon.

* [!!Really Major !!] the claims relative to change in the size / density / defective nature of the SiC NWs with varying welding temperature are unsubstantiated in the current state of the manuscript (line 280 and after).

- Detailed statistical analyses are required for size and density, possibly from SEM images

- Defect investigation using TEM are required for the defect quantification.

→ The above would require a very large body of experimental work to scientifically prove that the NW morphology or defect density are the cause of the bell curve of Fig14.

In my opinion, the authors should only state briefly that the origin of the increase and decrease could be linked to the NW morphology and density, removing all unsubstantiated sentence on NW density or defect, and stating that more detailed investigation are required to fully understand the process.

In that respect, the writing of the conclusion is scientifically correct “possibly ascribed to the bridging of nanowires in the joint.”

I recommend a "major" revision to fix the issue above.

Reviewer 2 Report

Discussion: 

Line 289. "the size of SiC nanowires increase gradually and become longer (as shown in Fig. 15f,g)" the use of the world size isn't precise if it means length or diameter. it many be written as "the length of SiC nanowires increase gradually (as shown in Fig. 15f,g)" 

Line 292. "nanowires coarsen and deform as the temperature continues to rise (as shown in Fig. 15h). This may cause more defects to form inside the nanowires" . I believe you got the cause and effect backwards. At higher temperature thermodynamically SIC will start favoring other polytypes such as 4H, 6H due to this stacking faults will be more prevalent and hence this results in misshapen, mechanically weak nanowire.  In VLS growth NW diameter(Coarseness) strictly relies on diameter of the catalyst droplet. The catalyst diameter decrease with increase in temperature but with decreased viscosity of the catalyst droplets sometimes the catalyst flow more readily combining forming large catalyst and hence larger NW.

I would suggest trying to do a post anneal at a higher temperature  and observe how the strength changes.

Figure:

Fig 9:  It is a good practice to insert scale bar in diffraction pattern. 

Fig.10:  10a image is not clear, 10b should show more than just four point in one direction alone to ascertain it is infact 3C-SiC( your Fig 9 is a excellent example of diffraction pattern). Also 10b should have a scale bar in reciprocal space.  If unable to acquire a better SAED for any reason, presence of too many stacking faults, spot size, background noise etc. you can measure d spacing of  other planes in the HRTEM image and look at the angle between the planes to confirm it is infact 3C-SiC. Fig 10b doesn't give readers any information.

Conclusion:

1)"The results showed that no distinct phase change in the porous BN/Si3N4 ceramic joint." The statement should be made clear. The whole point of the paper is the creation of SiC NW in the joint improving the joint strength. It may be modified to " The results showed no distinct phase change to the porous BN/Si3N4 part of the ceramic joint." or "The results showed that no distinct phase change in the porous BN/Si3N4 ceramic joint"

when you say "Compared with the uncoated porous BN/Si3N4 ceramic joint, the highest joint strength of the carbon-coated BN/Si3N4 joint 324 was ~89 MPa.." you should also specify the reference joint strength which is joint strength of uncoated porous BN/Si3N4 ceramic joint or you can say the same has increase by __%.

Round 2

Reviewer 1 Report

The authors have successfully replied to all the remarks. My recommendation is to accept the manuscript

Reviewer 2 Report

The revisions made by the author is satisfactory and is recommended for publication.